# PPT: TOKEN PRUNING AND POOLING FOR EFFICIENT VISION TRANSFORMERS

## ABSTRACT

Vision Transformers (ViTs) have emerged as powerful models in the field of computer vision, delivering superior performance across various vision tasks. However, the high computational complexity poses a significant barrier to their practical applications in real-world scenarios. Motivated by the fact that not all tokens contribute equally to the final predictions and fewer tokens bring less computational cost, reducing redundant tokens has become a prevailing paradigm for accelerating vision transformers. However, we argue that it is not optimal to either only reduce inattentive redundancy by token pruning, or only reduce duplicative redundancy by token merging. To this end, in this paper we propose a novel acceleration framework, namely token Pruning & Pooling Transformers (PPT), to adaptively tackle these two types of redundancy in different layers. By heuristically integrating both token pruning and token pooling techniques in ViTs without additional trainable parameters, PPT effectively reduces the model complexity while maintaining its predictive accuracy. For example, PPT reduces over 37% FLOPs and improves the throughput by over 45% for DeiT-S without any accuracy drop on the ImageNet dataset.

## 1 INTRODUCTION

In recent years, vision transformers (ViTs) have demonstrated promising results in many vision tasks such as image classification (Dosovitskiy et al., 2021; Jiang et al., 2021; Touvron et al., 2021; Yuan et al., 2021), object detection (Carion et al., 2020; Dai et al., 2021; Li et al., 2022; Zhu et al., 2021), and semantic segmentation (Kirillov et al., 2023; Liu et al., 2021; Wang et al., 2021). Compared with the convolutional neural networks (CNNs), ViTs have the property of modeling long-range dependencies with the attention mechanism (Vaswani et al., 2017), which introduces fewer inductive biases hence has the potential to absorb more training data. However, densely modeling long-range dependencies among image tokens can lead to computational inefficiency, especially when dealing with large datasets and training iterations (Carion et al., 2020; Dosovitskiy et al., 2021). This in turn limits further implementation of ViTs in real-world scenarios.

Given the strong correlation between model complexity and the number of tokens in ViTs, a direct approach to accelerate ViTs is to reduce the number of redundant tokens. Moreover, some studies also showed that not all tokens contribute equally to the final predictions (Caron et al., 2021; Pan et al., 2021) . Existing attempts to achieve token compression mainly consist of two branches of solutions, *i.e.*, token pruning and token pooling. The former approach emphasizes the design of different importance evaluation strategies to identify and retain relevant tokens while discarding irrelevant ones, as demonstrated in previous research (Fayyaz et al., 2022; Liang et al., 2022; Rao et al., 2021; Xu et al., 2022; Tang et al., 2022). The latter technique primarily concentrates on merging similar image tokens using a predefined similarity evaluation metric and merge policy (Bolya et al., 2023; Marin et al., 2021). To summarize, there are two types of redundancy in vision transformers, *i.e.*, **inattentive redundancy** and **duplicative redundancy**. However, it should be noted that the aforementioned each method only addresses one type of redundancy, *i.e.*, token pruning for inattentive redundancy while token pooling for duplicative redundancy. We argue that reducing only one type of redundancy leads to suboptimal acceleration performance.

In this paper, we propose a novel framework, named as token Pruning & Pooling Transformers (PPT), to jointly tackle the two types of redundancy as shown in Figure 1. Our research investigates that the importance of image tokens becomes more distinct as the layer deepens, indicating that applying token pruning techniques at deeper layers is more suitable. In contrast, the model prefers to use token pooling methods in shallow layers, where a large number of tokens exhibit relatively high similarity. Additionally, we observe that the distribution of token scores varies among different samples within the same layer. Therefore, we propose an instance-aware adaptive strategy to automatically choose optimal policy of token pruning or pooling in different layers. Moreover, our method introduces *no trainable parameters*. This indicates that it can

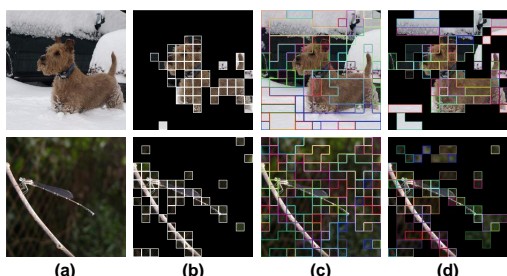

Figure 1: Visualizations of token compression results using different methods on the ImageNet dataset with DeiT-S. (a) Original images. (b) Token pruning methods, which discard inattentive tokens. (c) Token pooling methods, which merge similar tokens within the same color bounding box. (d) Our method can effectively address both types of redundancy while achieving superior performance.

be easily integrated into pre-trained ViTs with minimal accuracy degradation, and fine-tuning with PPT can lead to improved accuracy and faster training speed, making it especially beneficial for huge models. Our method is extensively evaluated for different benchmark vision transformers on ImageNet dataset. Experiment results demonstrate that our method outperforms state-of-the-art token compression methods, and achieves superior trade-off between accuracy and computational.

We summarize the main contributions as follows: (1) We propose PPT, which is a heuristic framework that acknowledges the complementary potential of token pruning and token merging techniques for effifient ViTs. (2) We design a redundancy criterion to guide adaptive decision-making on prioritizing different token compression policies for various layers and instances. (3) Our method is simple yet effective and can be easily incorporated into the standard transformer block *without additional trainable parameters*. (4) We perform extensive experiments and obtatin promising results for several different ViTs, *e.g.*, PPT can **reduce over 37% FLOPs** and **improve the throughput by over 45%** for DeiT-S **without any accuracy drop** on ImageNet. We hope our PPT could bring a new perspective for obtaining efficient vision transformers.

## 2 RELATED WORK

**Vision Transformers.** Inspired by the success of Transformers in natural language processing area (Brown et al., 2020; Kenton & Toutanova, 2019; Vaswani et al., 2017), the recent advance ViT (Dosovitskiy et al., 2021) shows that the transformer can also achieve promising results in the computer vision field. However, ViT requires large-scale datasets such as ImageNet-22K and JFT-300M for model pretraining and huge computation resources. Later, DeiT (Dosovitskiy et al., 2021) addressed this issue by optimizing the training strategy and introducing a distilled token, which is designed to learn knowledge from the teacher network and improve model performance. Some following works (Chen et al., 2021; Han et al., 2021; Liu et al., 2021; Wang et al., 2021; Wu et al., 2021; Yuan et al., 2021) focus on modifying patch embedding or transformer blocks to introduce local dependencies into ViTs, resulting in significant performance improvements. LV-ViT (Jiang et al., 2021) further improves ViTs by utilizing the local information embedded in patch tokens with a new training objective called token labeling. Although the ViT and its follow-ups have achieved excellent performance and demonstrated their strong potential as an alternative to CNNs, the high computational costs remain a challenge for practical implementation in real-world scenarios. The computational cost of ViTs is mainly attributed to the quadratic and linear time complexity of the multi-head self-attention (MSA) and feed-forward network (FFN), respectively, with respect to the number of tokens. As a result, researchers have been exploring token compression as a prevalent paradigm for accelerating ViTs, which can effectively alleviate the computational burden while retaining their expressive power and performance.

**Token Pruning.** As a main branch of token compression, token pruning aims to retain attentive tokens and prune inattentive ones by designing importance evaluation strategies. In Tang et al. (2022), The authors adopt a top-down paradigm to estimate the impact of each token and remove redundant

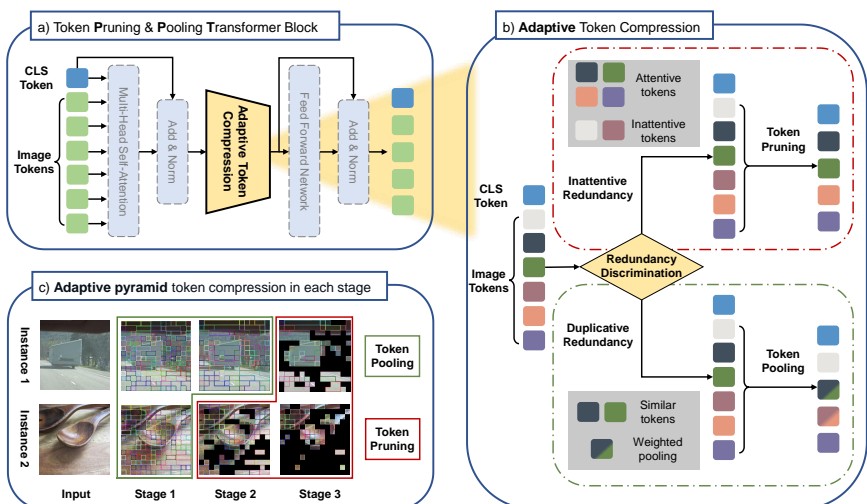

Figure 2: **Overview of the proposed PPT approach**. (a) The **Adaptive Token Compression** module is simple and can be easily inserted inside the standard transformer block *without additional trainable parameter*. (b) Our module can **adaptively** select either token pruning **or** token pooling policy to tackle corresponding redundancy based on the current token distribution, which is intuitively reflected across various instances and layers in (c). (c) With PPT, similar patches within the same color bounding box are pooled into a single token, while the masked inattentive patches are pruned, resulting in promising trade-offs between the accuracy and FLOPs.

ones. The IA-RED$^2$ (Pan et al., 2021) is designed to reduce input-uncorrelated tokens hierarchically, while also taking into account interpretability. DynamicViT (Rao et al., 2021) introduces lightweight prediction modules to score tokens and discard unimportant tokens. Evo-ViT (Xu et al., 2022) and EViT (Liang et al., 2022) show that the attention scores between classification tokens and image tokens can be utilized for importance assignment. A-ViT (Yin et al., 2022) and ATS (Fayyaz et al., 2022) go further by dynamically adjusting the pruning rate based on the complexity of the input image. However, mainstream deep learning frameworks do not fully support dynamic token-length inference during batch processing. While token pruning methods achieve promising performance, we realize that they pay less attention to redundancy in the foreground region.

**Token Pooling.** On the other hand, there are some attempts that recognize the significance of pooling tokens together. To alleviate information loss, Evo-ViT (Xu et al., 2022) and EViT (Liang et al., 2022) merge the tokens they pruned into a single token. To improve the efficiency of ViTs, *Token Pooling* (Marin et al., 2021) utilizes a K-Means-based clustering approach to exploit redundancies in the images, while ToMe (Bolya et al., 2023) thoroughly investigates token similarity and proposes a Bipartite Soft Matching algorithm (BSM) to gradually merge similar tokens. Token pooling methods can reduce the duplicative redundancy, but they do not take into account that not all tokens contribute equally to the final prediction.

# 3 METHODOLOGY

## 3.1 OVERVIEW

Compared to existing works, our method comprehensively takes into account two types of redundancy present in images, *i.e.*, inattentive redundancy and duplicative redundancy. To address these issues, we heuristically integrate both token pruning and token pooling techniques, resulting in favorable trade-offs between the accuracy and FLOPs of ViTs.

As shown in Figure 2, our approach can adaptively select either token pruning or token pooling policy based on the current token distribution, which is reflected across different inputs and layers. Furthermore, our method does not involve any trainable parameters, which makes it suitable *with or without training*. It can achieve impressive results even in *off-the-shelf* scenarios, where no customization or fine-tuning is required. In this section, we first introduce the token pruning and token pooling techniques utilized in our method (Section 3.2 and Section 3.3), and then describe how we integrate them to achieve adaptive token compression in detail (Section 3.4).

## 3.2 TOKEN PRUNING FOR INATTENTIVE REDUNDANCY

In general, the token pruning paradigm consists two steps, token scoring and token selecting. Token scoring assigns a score to each token based on its importance for the task, and then token selecting determines which tokens to keep and which to discard.

**Token Scoring.** There are quite a few non-parametric scoring mechanisms for tokens in prior work, *e.g.*, the attention scores $A_{cls}$ between the classification token and image tokens (Liang et al., 2022; Xu et al., 2022). Furthermore, ATS (Fayyaz et al., 2022) takes the values matrix $V$ into consideration and proposes a more comprehensive token scoring metric:

$$A = Softmax(\frac{Q \cdot K^T}{\sqrt{d}}), \quad \text{Score}_i = \frac{A_{1,i+1} \times \|V_{i+1}\|}{\sum_{j=1}^{N} A_{1,j+1} \times \|V_{j+1}\|}. \tag{1}$$

**Token Selecting.** For token selection, we return to the traditional Top-K selection policy for more controllable compression ratio, *i.e.*, preserve the Top-K important tokens while remove the other inattentive tokens, which is widely used in many works (Liang et al., 2022; Rao et al., 2021).

## 3.3 TOKEN POOLING FOR DUPLICATIVE REDUNDANCY

The token pooling techniques aims to merge similar image tokens together and can decrease the duplicative redundancy in the model. Recently, the Bipartite Soft Matching algorithm (BSM) algorithm (Bolya et al., 2023) shows superior performance in token merging. BSM first partitions the tokens into two sets of roughly equal size. It then draws an edge from each token in one set to the token in the other set with the highest *cosine* similarity score. The top-K-similar edges are selected, and tokens that are still connected are merged by averaging their features. In addition, it is necessary to maintain a variable $s$ that tracks the size of the tokens in order to minimize information loss. $s$ is a row vector that reflects the number of tokens represented and are combined with tokens any time. In addition, it is also used as a weight to reflect its importance in the calculation of attention matrix:

$$A = Softmax(\frac{Q \cdot K^T}{\sqrt{d}} + \log s). \tag{2}$$

We also explore another clustering-based token pooling method in Section C of appendix and demonstrate the BSM performer better in our framework.

## 3.4 TOKEN PRUNING & POOLING TRANSFORMER

It is a nontrivial idea to combine token pruning for inattentive tokens and token pooling for attentive tokens. For example, an intuitive approach is to utilize both techniques within the same block. However, we find that this simple strategy does not obtain satisfactory performance, as shown in Section 4.2. To dig deeper into the possible reason behind, we first perform a comprehensive analysis of token pruning and token pooling techniques and find out an interesting observation for the token redundancy of different layers. To this end, we propose an adaptive token compression method to automatically discover the best policies to deal with the two types of redundancy in different layers.

**A Closer Look at Token Pruning and Token Pooling.** We first conduct some analysis of the importance scores in different layers. Our research reveals an intriguing phenomenon: the variance of significance scores assigned to image tokens for each sample increases as the number of layers in the model increases, as depicted in Figure 3 (deeper analysis are shown in Section B of the appendix). This suggests that *the importance of image tokens becomes more distinct as the layer deepens*. As a result, *token pruning techniques may be preferred at deeper layers*, where certain tokens exhibit significantly lower importance scores and are therefore more likely to be pruned. Meanwhile, premature token pruning at shallow layers may result in irreversible information loss and negatively impact model performance.

In contrast, we find *the token pooling techniques are preferably applied in shallow layers* through our analysis. Since we use a pyramid compression approach for tokens, there are plenty of tokens in the shallow layers that exhibit relatively high similarity. This makes it less concerning to merge dissimilar tokens during token pooling under the Top-K strategy, as it is unlikely to affect the performance of the model. Moreover, since we maintain the vector $s$ that reflects the size of each token, the

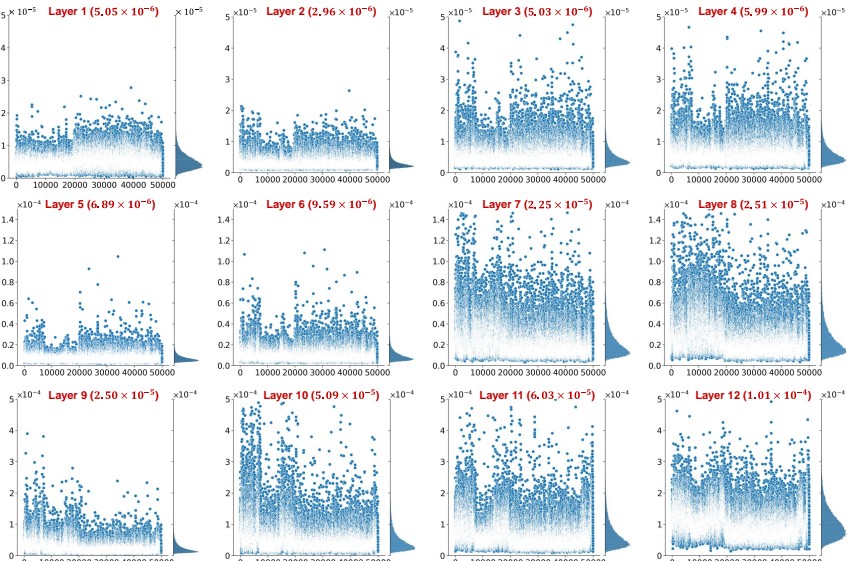

Figure 3: The scatter and the histogram of the variance of the significance scores assigned to image tokens at each layer of the DeiT-S model on the ImageNet validation set. The y-axis corresponds to the variance value and the x-axis to the index of samples in the dataset. We display the average variance of each layer at the top of each graph in red to track the trend of variance changes as the layers go deeper.

information loss and impact on model performance caused by merging highly similar tokens can be almost negligible. Additionally, due to the decreasing number of similar tokens, the utilization of token pooling techniques in deeper layers is not applicable.

**Adaptive Token Pruning & Pooling Strategy.** Based on the above discussion, we propose an adaptive token compression method to automatically discover the best policies for removing duplicative and inattentive redundancy. In this way, our method can adaptively select either token pruning **or** token pooling to tackle corresponding redundancy based on the current significance scores of tokens. In addition, as shown in Figure 3, we can observe that the variance of the significance scores assigned to image tokens *varies among different samples* within the same layer. This implies that defining token compression rules solely based on the layer is not sufficient. Therefore, we propose an adaptive strategy that takes into account both the instances and layers, as follows:

$$S_{op_i} = var(\text{Score}_i), \quad \text{Policy}_i = \begin{cases} \text{Token Pruning, if } S_{op_i} > \tau \\ \text{Token Pooling, otherwise} \end{cases}, \quad (3)$$

where the $\text{Score}_i$ is calculated in formula 1 and the $\tau$ is a hyperparameter, *i.e.*, decision threshold, that regulates the model's preference for either of the two strategies. The impact of $\tau$ on model performance is thoroughly explored in Section 4.2. Taking a holistic view of our approach, we find that *no additional learnable parameter* is introduced, which indicates that our method can be seamlessly integrated with pre-trained ViTs and yields competitive performance, as demonstrated by our experiment results.

## 4 EXPERIMENTS

In this section, we empirically investigate the superiority of the proposed PPT through extensive experiments on ImageNet-1K (ILSVRC2012) (Deng et al., 2009), which contains approximately 1.28M training images and 50K validation images. We compare our proposed model with state-of-the-art models and conduct thorough ablation studies to gain a better understanding of its effectiveness.

**Implementation details.** We conduct experiments on the standard ViTs (Dosovitskiy et al., 2021) (including DeiT-Ti, DeiT-S, DeiT-B (Touvron et al., 2021)) and the different variants of ViTs (such as LV-ViT-S (Jiang et al., 2021), T2T-ViT(Yuan et al., 2021), and PS-ViT(Yue et al., 2021)). Following Liang et al. (2022), we specifically introduce our method into the $4^{th}$, $7^{th}$, and $10^{th}$

Table 1: Comparison of the accelerated vision transformers with different methods applied to multiple vanilla ViTs on ImageNet. The model with '†' is trained from scratch (300 epochs).

| Model | Method | Top-1 Acc. (%) | Params (M) | FLOPs (G) | Throughput (image / s) |
|-------|--------|----------------|------------|-----------|------------------------|
| DeiT-Ti | Baseline (Touvron et al., 2021) | 72.2 | 5.6 | 1.3 | 2675 |
| | DynamicViT (Rao et al., 2021) | 71.4 (↓ 0.8) | 5.9 | 0.8 (↓ 38.5%) | 3765 (↑ 40.7%) |
| | Evo-ViT† (Xu et al., 2022) | 72.0 (↓ 0.2) | 5.9 | 0.8 (↓ 38.5%) | 3781 (↑ 41.3%) |
| | EViT (Liang et al., 2022) | 71.9 (↓ 0.3) | 5.6 | 0.8 (↓ 38.5%) | 3387 (↑ 26.6%) |
| | ToMe† (Bolya et al., 2023) | 71.4 (↓ 0.8) | 5.6 | 0.8 (↓ 38.5%) | 3685 (↑ 37.7%) |
| | **PPT (off-the-shelf) (Ours)** | 71.6 (↓ 0.6) | 5.6 | 0.8 (↓ 38.5%) | 3572 (↑ 33.5%) |
| | **PPT (Ours)** | **72.1 (↓ 0.1)** | 5.6 | 0.8 (↓ 38.5%) | 3572 (↑ 33.5%) |
| DeiT-S | Baseline (Touvron et al., 2021) | 79.8 | 22.1 | 4.6 | 993 |
| | DynamicViT (Rao et al., 2021) | 79.3 (↓ 0.5) | 22.8 | 3.0 (↓ 34.8%) | 1440 (↑ 45.0%) |
| | IA-RED² (Pan et al., 2021) | 79.1 (↓ 0.7) | - | 3.2 (↓ 30.4%) | 1362 (↑ 37.2%) |
| | PS-ViT (Tang et al., 2022) | 79.4 (↓ 0.4) | - | 2.6 (↓ 43.5%) | 1321 (↑ 33.0%) |
| | Evo-ViT† (Xu et al., 2022) | 79.4 (↓ 0.4) | 22.4 | 3.0 (↓ 34.8%) | 1414 (↑ 42.4%) |
| | EViT (Liang et al., 2022) | 79.5 (↓ 0.3) | 22.1 | 3.0 (↓ 34.8%) | 1378 (↑ 38.8%) |
| | ATS (Fayyaz et al., 2022) | 79.7 (↓ 0.1) | 22.1 | 2.9 (↓ 37.0%) | 1382(↑ 39.2%) |
| | ToMe† (Bolya et al., 2023) | 79.4 (↓ 0.4) | 22.1 | 2.7 (↓ 41.3%) | 1552 (↑ 56.3%) |
| | **PPT (off-the-shelf) (Ours)** | 79.5 (↓ 0.3) | 22.1 | 2.9 (↓ 37.0%) | 1448 (↑ 45.8%) |
| | **PPT (Ours)** | **79.8 (↓ 0.0)** | 22.1 | 2.9 (↓ 37.0%) | 1448 (↑ 45.8%) |
| DeiT-B | Baseline (Touvron et al., 2021) | 81.8 | 86.6 | 17.6 | 295 |
| | DynamicViT (Rao et al., 2021) | 81.3 (↓ 0.4) | 89.4 | 11.5 (↓ 34.6%) | 454 (↑ 53.8%) |
| | IA-RED² (Pan et al., 2021) | 80.3 (↓ 1.5) | - | 11.8 (↓ 33.0%) | 453 (↑ 53.6%) |
| | Evo-ViT† (Xu et al., 2022) | 81.3 (↓ 0.5) | 87.3 | 11.7 (↓ 33.5%) | 429 (↑ 45.4%) |
| | EViT (Liang et al., 2022) | 81.3 (↓ 0.5) | 86.6 | 11.6 (↓ 34.1%) | 440 (↑ 49.2%) |
| | **PPT (off-the-shelf) (Ours)** | 80.3 (↓ 1.5) | 86.6 | 11.6 (↓ 34.1%) | 445 (↑ 50.8%) |
| | **PPT (Ours)** | **81.4 (↓ 0.4)** | 86.6 | 11.6 (↓ 34.1%) | 445 (↑ 50.8%) |

layers of DeiT models and into the $5^{th}$, $9^{th}$, and $13^{th}$ layers of LV-ViT models. For all comparative experiments, we report the performance of our method in terms of both *off-the-shelf* and *fine-tuned*. Following the approach in Rao et al. (2021), we initialize the backbone models with official pre-trained ViTs. If fine-tuning is applied, we jointly train the entire models for 30 epochs, similar to other works (Fayyaz et al., 2022; Liang et al., 2022; Rao et al., 2021). Regarding training strategies and optimization methods, we follow the setup described in the original papers of DeiT (Touvron et al., 2021) and LV-ViT (Jiang et al., 2021), except that the basic learning rates are set to be $\frac{batchsize}{128} \times 10^{-5}$. The image resolution used for both training and testing is $224 \times 224$. The decision threshold $\tau$ we introduced in our framework is $7 \times 10^{-5}$ and $5 \times 10^{-4}$ in DeiT and LV-ViT-S, respectively. All the experiments are conducted using PyTorch on NVIDIA GPUs. We report the top-1 classification accuracy and floating-point operations (FLOPs) to evaluate model efficiency. Additionally, we measure the throughput of the models on a single NVIDIA V100 GPU with batch size fixed to 256 same as Tang et al. (2022); Xu et al. (2022).

## 4.1 MAIN RESULTS

**Comparisons with existing token compression methods.** Even though we do not add extra parameters as Rao et al. (2021), or apply complicated token reorganization tricks as Liang et al. (2022); Xu et al. (2022), the experimental results, as presented in Table 1 and Figure 4, demonstrate that our method achieves higher accuracy with comparable computation cost. Specifically, PPT reduces FLOPs by over 37% and improves throughput by over 45% without any accuracy drop on the classic DeiT-S model as shown in Table 1. Furthermore, the superiority of PPT is evident across various FLOPs when compared with other token compression methods, as shown in Figure 4. Specifically, our method excels in the following three aspects: Firstly, at lower FLOPs (below 2.5G), our method showcases a noteworthy performance improvement of 0.7%-3.8% compared to other methods. Secondly, at higher FLOPs (3.0G above), our method outperforms the baseline model and surpasses the capabilities of the comparison methods. Lastly, when used in an off-the-shelf setting, PPT demonstrates significant improvement of 0.1%-2.3% across various FLOPs compared to other methods, achieving competitive results with those obtained by fine-tuned models.

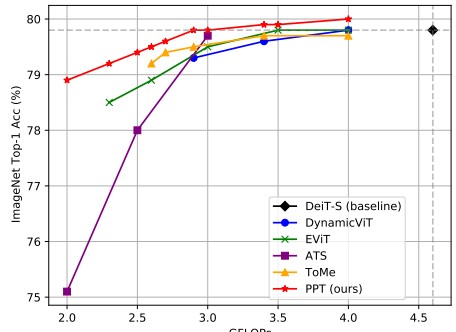 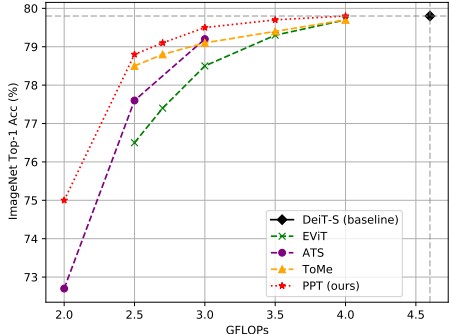

Figure 4: **Comparison between our method and other methods under different FLOPs**. We conducted a comprehensive comparison of the performance of various methods after **fine-tuning (left)** and **off-the-shelf (right)**, which highlights the superior performance of our method.

Table 2: Comparisons with different variants of ViTs on ImageNet. We compress the LV-ViT-S (Jiang et al., 2021) as the base model and achieve promising accuracy-FLOPs trade-off.

| Model | Params (M) | FLOPs (G) | Top-1 Acc. (%) |
|---|---|---|---|
| DeiT-S | 22.1 | 4.6 | 79.8 |
| DeiT-B | 86.6 | 17.6 | 81.8 |
| PVT-Small | 24.5 | 3.8 | 79.8 |
| PVT-Medium | 44.2 | 6.7 | 81.2 |
| CoaT-Lite Small | 20.0 | 4.0 | 81.9 |
| CrossViT-S | 26.7 | 5.6 | 81.0 |
| Swin-T | 29.0 | 4.5 | 81.3 |
| Swin-S | 50.0 | 8.7 | 83.0 |
| T2T-ViT-14 | 22.0 | 4.8 | 81.5 |
| T2T-ViT-24 | 64.1 | 14.1 | 82.3 |
| RegNetY-8G | 39.0 | 8.0 | 81.7 |
| RegNetY-16G | 84.0 | 16.0 | 82.9 |
| PS-ViT-B/14 | 21.3 | 5.4 | 81.5 |
| PS-ViT-B/18 | 21.3 | 8.8 | 82.3 |
| PiT-S | 23.5 | 2.9 | 80.9 |
| PiT-B | 73.8 | 12.5 | 82.0 |
| CvT-13 | 20.0 | 4.5 | 81.6 |
| TNT-S | 23.8 | 5.2 | 81.5 |
| TNT-B | 66.0 | 14.1 | 82.9 |
| LV-ViT-S | 26.2 | 6.6 | 83.3 |
| DynamicViT-LV-S | 26.9 | 4.6 | 83.0 |
| PS-LV-ViT-S | 26.2 | 4.7 | 82.4 |
| EViT-LV-S | 26.2 | 4.7 | 83.0 |
| **PPT-LV-S (Ours) (off-the-shelf)** | 25.8 | 4.6 | 82.8 |
| **PPT-LV-S (Ours)** | **25.8** | **4.6** | **83.1** |

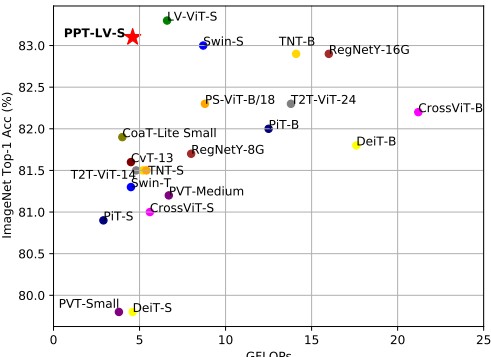

Figure 5: Comparison of different models with various accuracy-FLOPs trade-off. Our PPT-LV-S achieves a quite competitive trade-off than other ViTs.

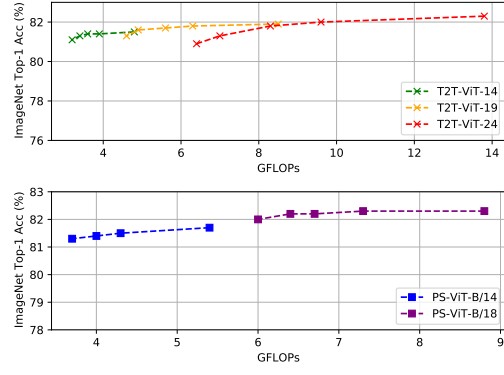

Figure 6: The performance of applying our method to more ViTs (without fine-tuning).

**Application on different variants of ViTs.** Apart from the standard ViTs (Dosovitskiy et al., 2021; Touvron et al., 2021), subsequent studies (Chen et al., 2021; Han et al., 2021; Heo et al., 2021; Jiang et al., 2021; Liu et al., 2021; Radosavovic et al., 2020; Wang et al., 2021; Wu et al., 2021; Xu et al., 2021; Yuan et al., 2021; Yue et al., 2021) further improve the performance of ViTs by varying the initial architecture or optimization strategies. To further showcase the potential and superiority of our approach, we extend the integration of PPT with the LV-ViT-S. As shown in Table 2 and Figure 5, our PPT-LV-S can achieve a highly competitive performance among numerous vision transformers from the perspective of accuracy-computation trade-off, and further demonstrate superiority over other token compression methods on LV-ViT-S. To further demonstrate the generality and superiority of our approach, we also apply PPT on more variants of ViTs, such as T2T-ViT and PS-ViT, as shown in Figure 6. More detailed results are shown in Section B of the appendix.

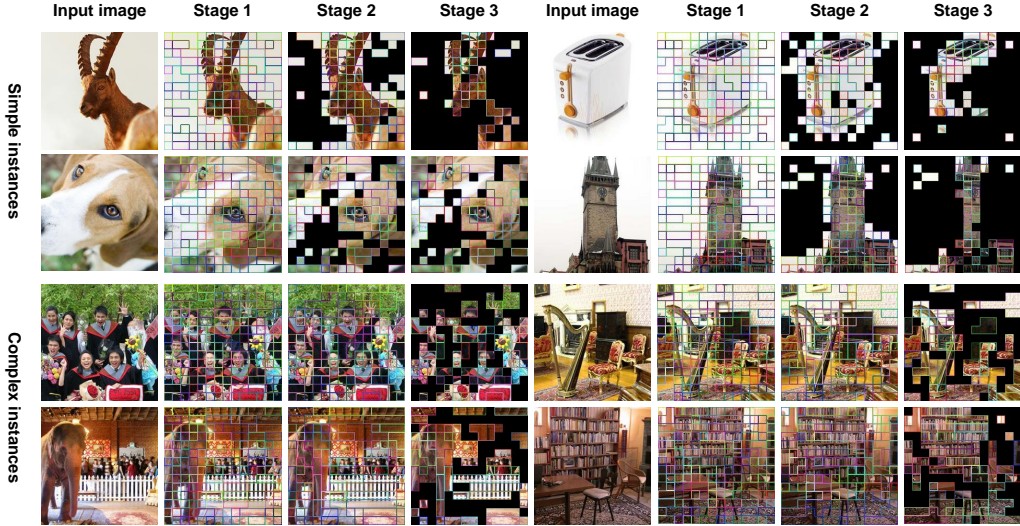

Figure 7: **Visualizations of token compression results on DeiT-S**. The masked regions represent the inattentive redundancy and are pruned, while the patches with the same inner and border color means the duplicative redundancy and are pooled. The redundancy in the image is pyramid reduced, and our method can adaptively execute different strategies at different stages based on the input. We demonstrate the generalization of the model to images of varying complexity. More results are visualized in the Section D of the appendix.

**Visualizations of token compression results.** To gain further insight into the interpretability of PPT, we conducted a visualization analysis of the intermediate process of our token compression in Figure 7. As expected, we observe that PPT tends to implement token pooling in shallow layers and token pruning in deep layers, and leverage incompletely consistent compression strategies for different images. As the network deepens, the duplicative redundancy and inattentive redundancy gradually removed, while the most informative tokens are reserved. From the final output, we observed that PPT can assist ViTs in focusing on patches specific to the target class, and we also demonstrated that background patches can be meaningful for recognition. The visualizations corroborate that our approach is effective in processing images irrespective of whether the backgrounds are simple or complex. And the results demonstrate that the model is more cautious when processing complex and information-rich images, preferring token pooling strategies with lower information loss. Conversely, for simpler images, the model is more decisive and tends to use token pruning strategies. These observations underscore the importance of our adaptive token compression strategy intuitively.

### 4.2 ABLATION STUDY

We conduct extensive ablation studies to explore the effectiveness of each component in our method. The DeiT-S is used as the default model.

**Effectiveness of techniques integration.** First, we analyze the effectiveness of our proposed framework by exploring the performance of using token pruning or token pooling separately, and a comparatively trivial integration approach, *i.e.*, naively combining token pruning and pooling within a block, under different FLOPs settings. As illustrated in Figure 8, when only using one technique, token pooling performed better than token pruning *in our framework*. Combining the two techniques within a block demonstrates the advantage of techniques integration at lower FLOPs, but it may reduce the performance of token pooling at higher FLOPs. In contrast, our framework achieves optimal performance under different Flops, demonstrating its effectiveness.

**Different policy selection mechanisms.** We believe that different images possess varying levels of complexity and exhibit different types of redundancy. Therefore, dynamically selecting strategies based on the input can enhance the model's flexibility, as demonstrated in Figure 3 and Figure 7. To further demonstrate the necessity of this strategy, we conduct additional experiments (under an off-the-shelf scenario) to compare our adaptive selection strategy with the random policy and the rule-based policy. The rule-based policy involved utilizing token pooling for the initial half of

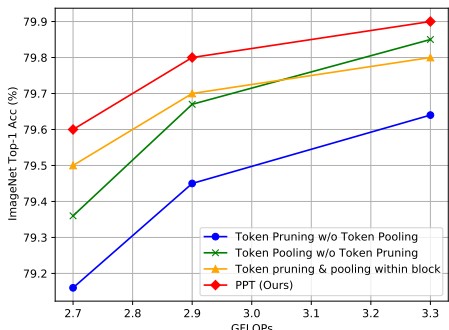
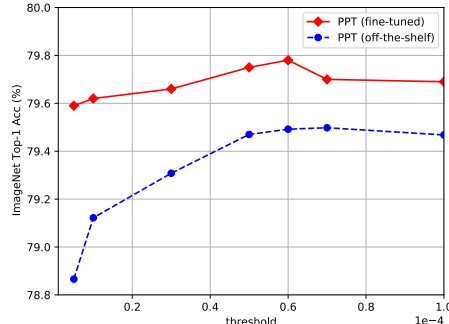

Figure 8: Comparison of the PPT framework with the individual modules and an alternative combination method under different FLOPs.

Figure 9: Impact of the decision threshold $\tau$ on the performance with or without training, using the value range referenced in the statistical data from Figure 3.

Table 3: Effectiveness of different policy selection mechanism.

| FLOPs (G) | Top-1 Acc(%) of various mechanisms | | | |
|---|---|---|---|---|
| | Random | Rule-Based | **Adaptive (Ours)** | Policy Inversion |
| 2.5 | 77.5 | 78.4 | **78.8** | 76.5 |
| 2.7 | 78.2 | 78.8 | **79.1** | 77.6 |
| 2.9 | 79.0 | 79.2 | **79.5** | 78.6 |

blocks and transitioning to token pruning for the remaining ones. Additionally, we explored the inversion of our policy decision, wherein pruning was performed instead of pooling, and vice versa. This exploration aimed to reinforce our claims that token pooling techniques are typically applied in shallow layers, while token pruning techniques are preferably employed in deeper layers. The detailed experimental results are presented in Table 3.

**Impact of the decision threshold.** As a key hyperparameter in PPT, the decision threshold $\tau$ controls the model's preference for token pruning and token pooling techniques. When the $\tau$ is a smaller value, the model is more likely to use the token pruning policy, and vice versa. It is important to set a optimal $\tau$ to balance the two strategies. Motivated by the statistical data in Figure 3, we explore the $\tau$ range from the average variance of first layer ($5 \times 10^{-6}$) to the last layer ($1 \times 10^{-4}$) and show the results in Figure 9. We observe the optimal $\tau$ is $6 \times 10^{-5}$ with fine-tuning and $7 \times 10^{-5}$ under *off-the-shelf*, respectively.

We also explore the **impact of different pruning and pooling policy** and **different metrics for redundancy discrimination** applied in our framework, we show the results in Section C due to the space limitations.

## 5 CONCLUSION

In this work, we bring a new perspective for obtaining efficient vision transformers by integrating both token pruning and token pooling techniques. Our proposed framework, named token Pruning & Pooling Transformers (PPT), adaptively decides different token compression policies for various layers and instances. The proposed method is simple yet effective and can be easily incorporated into the standard transformer block without additional trainable parameters. Extensive experiments demonstrate the effectiveness of our method. Specifically, our PPT can reduce over 37% FLOPs and improve the throughput by over 45% for DeiT-S without any accuracy drop on ImageNet. Despite the promising results, there are still some limitations to our proposed method that should be acknowledged. Specifically, PPT can not be directly applied to dense prediction tasks and self-supervised learning, because the classification token plays an important role in our framework.

Overall, our work offers a valuable contribution to the development of efficient vision transformers, highlighting the importance of adaptive token compression and providing new insights into the integration of token pruning and token pooling techniques. We hope that our method inspires new research and leads to further improvements in the field of efficient transformer-based models.

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

# A  DEEPER ANALYSIS

**Distributions of significance scores.** In Figure 10, we further investigate the impact of our method on the distribution of significance scores across different layers. Specifically, we introduce our method into the $4^{th}$, $7^{th}$, and $10^{th}$ layers of the DeiT-S model. As shown in comparison to the original model (Figure 3 in the main text), the first four layers remain unaffected, while the introduction of PPT in the fourth layer leads to a significant increase in variance for the subsequent layers. This phenomenon indicates that *our method makes the importance of image tokens more distinct*, which may be the underlying reason for the performance improvement achieved by our method.

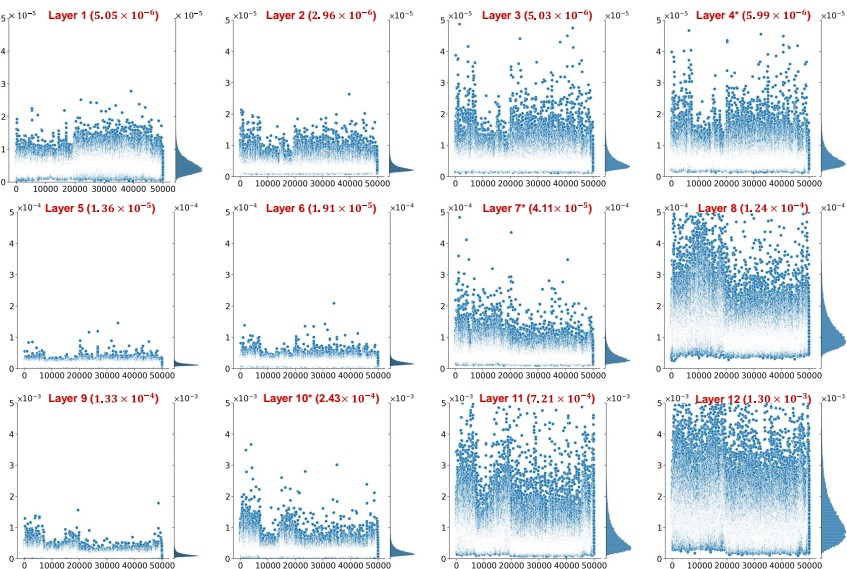

Figure 10: The scatter and the histogram of the variance of the significance scores assigned to image tokens at each layer of the **compressed** DeiT-S model on the ImageNet validation set, t**he layer with '\*' indicate the PPT block is inserted**. The y-axis corresponds to the variance value and the x-axis to the index of samples in the dataset. We display the average variance of each layer at the top of each graph.

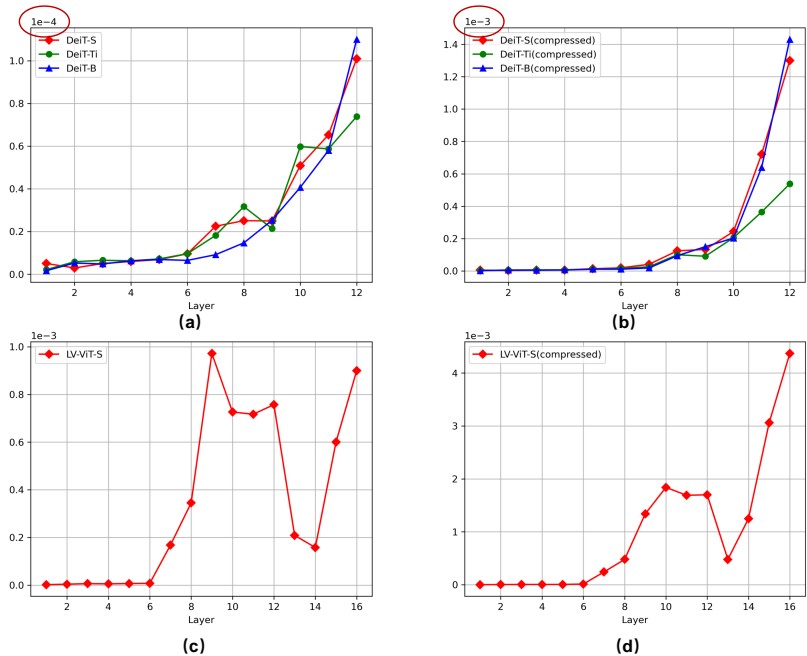

Figure 11: **The average variance of the significance scores at each layer** of different ViTs on the ImageNet validation set. Figure (a) and Figure (c) are generated based on the original models, while Figure (b) and Figure (d) are generated based on the compressed models with our method.

In Figure 11, we further refine our analysis by computing the average variance of significance scores across layers for different ViTs. We can observe a similar phenomenon across different ViTs, where ***the variance increases with layer depth, and compressed ViTs exhibit greater variance.*** This phenomenon to some extent validates the effectiveness and generality of our method.

# B  FULL RESULTS

We provide more detailed performance results of PPT on DeiT and LV-ViT-S in Table 4. Again, the superiority of PPT is evident under different FLOPs. Specifically, PPT maintains remarkably high accuracy even at higher compression rates, and interestingly, it even outperforms the original model at lower compression rates.

Table 4: Results of PPT on different ViTs under various FLOPs, ACC$^*$ indicates that the ACC is evaluated under off-the-shelf (without fine-tuning).

| Model | Removed tokens per stage | FLOPs (G) | Top-1 Acc. (%) | Top-1 Acc$^*$. (%) |
|---|---|---|---|---|
| ViT (DeiT)-Ti | 0 | 1.3 | 72.2 | 72.2 |
| | 10 | 1.16 | 72.32 | 72.09 |
| | 20 | 1.07 | 72.26 | 72.05 |
| | 30 | 0.97 | 72.25 | 72.06 |
| | 40 | 0.89 | 72.10 | 71.74 |
| | 45 | 0.84 | 71.99 | 71.61 |
| | 50 | 0.80 | 71.90 | 71.28 |
| | 60 | 0.74 | 71.58 | 70.54 |
| ViT (DeiT)-S | 0 | 4.6 | 79.8 | 79.8 |
| | 10 | 4.26 | 79.99 | 79.79 |
| | 20 | 3.92 | 80.00 | 79.81 |
| | 30 | 3.59 | 79.94 | 79.72 |
| | 40 | 3.26 | 79.89 | 79.65 |
| | 50 | 2.94 | 79.76 | 79.49 |
| | 60 | 2.72 | 79.58 | 79.13 |
| ViT (DeiT)-B | 0 | 17.6 | 81.8 | 81.8 |
| | 40 | 12.48 | 81.47 | 80.88 |
| | 47 | 11.60 | 81.37 | 80.30 |
| | 50 | 11.27 | 81.19 | 80.04 |
| LV-ViT-S | 0 | 6.6 | 83.3 | 83.3 |
| | 40 | 4.94 | 83.25 | 83.03 |
| | 50 | 4.60 | 83.09 | 82.82 |
| | 60 | 4.33 | 82.82 | 82.57 |

In addition, we provide the detailed data for plots in Figure 4 in Table 5 and Table 6.

Table 5: The corresponding data for Figure. 4 (a)

| FLOPs (G) | Top-1 Acc(%) of Models (Fine-tunned) | | | | |
|---|---|---|---|---|---|
| | DynamicViT | EViT | ATS | ToMe | **PPT (Ours)** |
| 2.0 | - | - | 75.1 | - | **78.9** |
| 2.3 | - | 78.5 | - | - | **79.2** |
| 2.5 | - | - | 78.0 | - | **79.4** |
| 2.6 | - | 78.9 | - | 79.2 | **79.5** |
| 2.7 | - | - | - | 79.4 | **79.6** |
| 2.9 | 79.3 | - | - | 79.5 | **79.8** |
| 3.0 | - | 79.5 | 79.7 | - | **79.8** |
| 3.4 | 79.6 | - | - | 79.7 | **79.9** |
| 3.5 | - | 79.8 | - | - | **79.9** |
| 4.0 | 79.8 | 79.8 | - | 79.7 | **80.0** |

Table 6: The corresponding data for Figure. 4 (b)

| FLOPs (G) | Top-1 Acc(%) of Models (Off-the-shelf) | | | |
|---|---|---|---|---|
| | EViT | ATS | ToMe | **PPT(Ours)** |
| 2.0 | - | 72.7 | - | **75.0** |
| 2.5 | 76.5 | 77.6 | - | **78.8** |
| 2.7 | 77.4 | - | 78.8 | **79.1** |
| 3.0 | 78.5 | 79.2 | 79.1 | **79.5** |
| 3.5 | 79.3 | - | 79.4 | **79.7** |
| 4.0 | 79.7 | - | 79.7 | **79.8** |

## C  EXTEND EXPERIMENTS

**Different token pruning policy.** As described in Section 3.2, whether to take values matrix $V$ into consideration influences the performance of the model. We furthermore explore the impact of different token scoring mechanisms comprehensively. As illustrated in Figure 12, we observe that the performance of the two methods is closely comparable under *off-the-shelf* while scoring with the norm of $V$ achieves slightly better results after fine-tuning.

**Different token pooling policy.** Here we discuss different design choices for token pooling. For example, we could also utilize the efficient density peak clustering algorithm (DPC) (Rodriguez & Laio, 2014) for token merging. It identifies density peaks based on the local density and distance between image tokens, and assigns each token to the nearest density peak and forms clusters by merging nearby tokens that belong to the same peak. Figure 13 presents the effects of different token pooling policies. Our experiments demonstrate that BSM can improve the results of DPC by approximately 0.1% with fine-tuning, and the advantage gap is even larger without training.

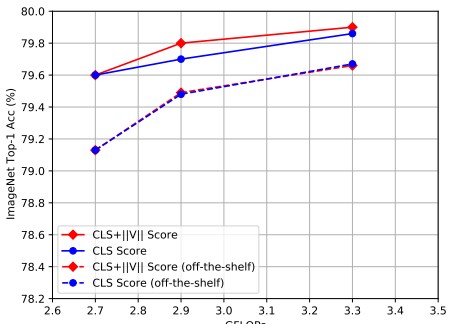

Figure 12: Impact of different score assignment methods when token pruning applied in our framework.

Figure 13: BSM *V.S.* DPC when token pooling applied in our framework.

**Different metrics for redundancy discrimination.** The reason we use the variance of the significant score as our policy score in our approach is because the variance reflects the dispersion of the values. A larger variance indicates a clearer distinction between important and unimportant image tokens, which is beneficial for token pruning. Conversely, when the variance is smaller, token pooling is more suitable. Additionally, we have explored another metric, the average of token similarity, which also reflects the redundancy level of tokens in different layers. However, through ablation study (without fine-tuning), we have observed that its performance is inferior to our chosen metric.

Table 7: The performance of different metrics.

| Metrics | GFLOPs | Top-1 ACC (%) |
|---|---|---|
| average of token similarity | 2.9 | 79.2 |
| variance of the significant score | | 79.5 |

*Typo correction*

## D   MORE VISUALIZATION

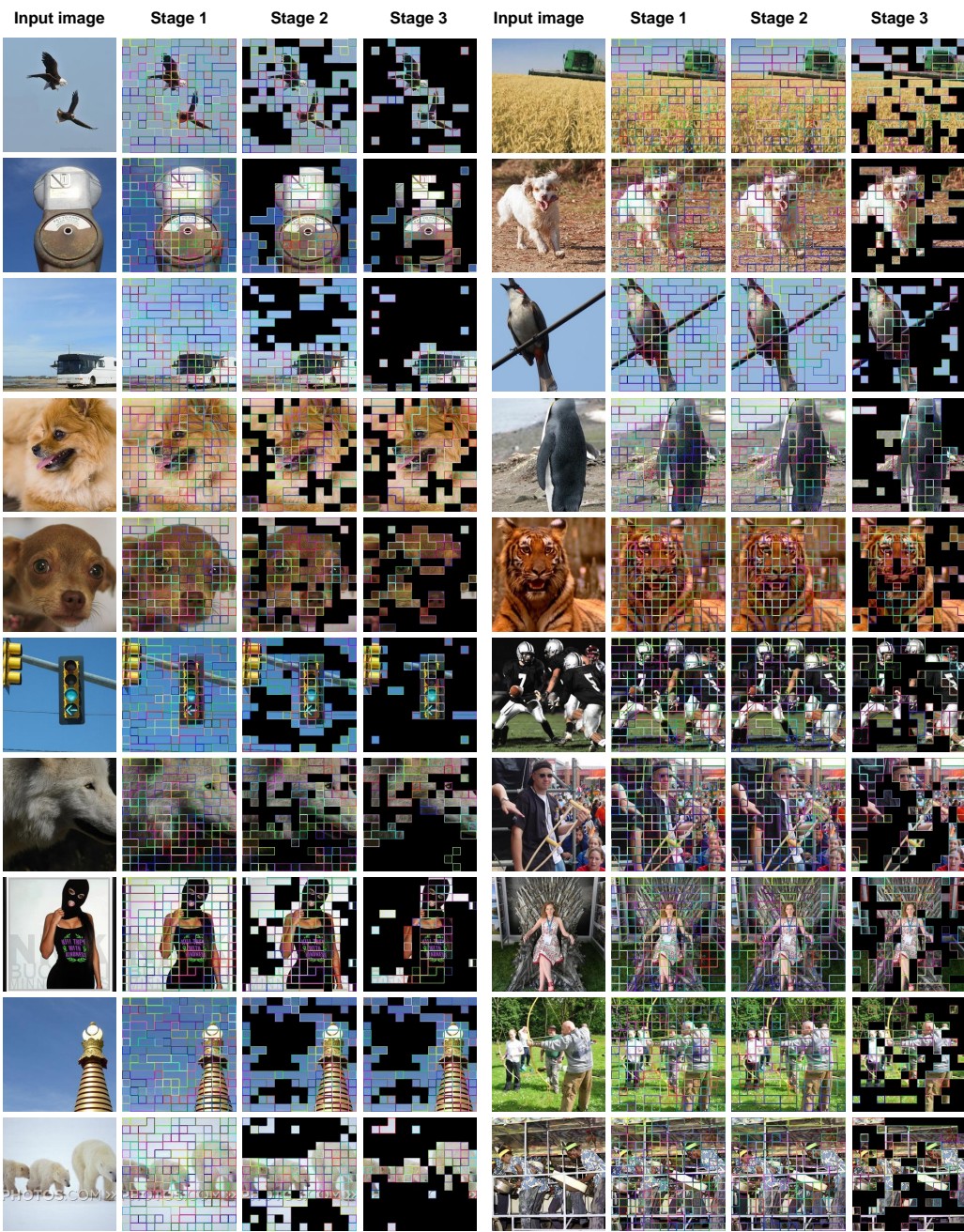

Figure 14: **Extended visualizations of our token compression results on DeiT-S with 12 layers**. The input image is sampled from the validation of ImageNet. The masked regions represent the inattentive redundancy and are pruned, while the patches with the same inner and border color means the duplicative redundancy and are pooled. The redundancy in the image is pyramid reduced, and our method can adaptively execute different strategies at different stages based on the input. We demonstrate our method works well for different images from various categories with various complexity.

