# OpenReview forum: "PPT: Token Pruning and Pooling for Efficient Vision Transformers"
_ICLR.cc/2024/Conference — ICLR 2024 Conference Withdrawn Submission_

### Official Review · Reviewer_b5bi · 2023-10-27

**Soundness:** 3 good
**Presentation:** 3 good
**Contribution:** 2 fair
**Rating:** 5
**Confidence:** 4

**Summary:**

This paper introduces a dynamic method that integrates token pruning and pooling (merging) to reduce the FLOPs and execution time of vision transformers. The authors note that as the layers become deeper, the significance of image tokens grows increasingly distinct. Consequently, applying token pruning in deeper layers and pooling in shallow ones should yield benefits. This adaptable strategy can function effectively without modification and delivers enhanced performance after fine-tuning. The central contribution of the paper lies in its validation of this adaptive approach, which effectively addresses inattentive redundancy and duplicative redundancy across different layers.

**Strengths:**

- The motivation for redundancy and duplicative redundancy should be handled differently across different layers is clear.
- Well-written and easy to follow.
- This method can work off-the-shell without finetuning.

**Weaknesses:**

- Lack of discussion and comparison with the highly related work "joint Token Pruning & Squeezing (TPS) "[1], whose method is doing the pruning and pooling(squeezing/merging) at the same time in a non-adaptive way, which should be a good and important baseline for this adaptive strategy.
- In the small network series such as Deit-S, the performance is 0.3% behind TPS[1] in a comparable FLOPS budget, I think the performance should be higher to justify the benefit of the adaptive strategy. It would be nice if the author would give more analysis about this case.
- Lack of experiments of hybrid ViT such as PVT[2]. I think it would make the method more impactful if it is applicable to hybrid ViT(include regular spatial conv) beyond vanilla ViTs.

[1] https://arxiv.org/abs/2304.10716
[2] https://arxiv.org/abs/2102.12122

**Questions:**

- The adaptive pruning and pooling strategy is the key contribution of this method. I think the author should make the "adaptive" point more convincing, such as comparing other non-adaptive pruning and pooling methods(pruning and pooling at the same time), or comparing the naive non-adaptive baseline(pruning only before layer x and pooling only after layer x, layer x is a fixed parameter such as half or two-thirds of the depth.
- I think more metrics other than the "variance" of the distribution of importance score of image tokens should be considered. Since variance is shift-invariant and absolute value of the score is not considered. The absolute value of the score distribution should be somehow helpful to the choice of pooling or merging.
- See weakness 3, how to make this method work on hybrid ViT(including regular spatial conv) and what's the improvement?
- How to make this method work on dense prediction tasks such as segmentation and what's the performance?
- Is it a possible case where pooling at the early stage and pruning at the final stage?
- What's the performance if the adaptive policy is reversed? (change pooling to pruning and pruning to pooling)

[1] https://arxiv.org/abs/2304.10716

---

> ### Author Response · Authors · 2023-11-11
> **Response to reviewer b5bi**
>
> We sincerely appreciate your thoughtful comments, efforts, and time. We respond to each of your concerns and questions one-by-one in what follows:
>
> > **"Lack of discussion and comparison with the highly related work "joint Token Pruning & Squeezing (TPS)..."**
>
> Thank you for the complement to our comparative methods, which is a great work. we provide additional comparative results based on the DeiT-S baseline as shown in the table below:
>
> |Model|Method|FLOPs (G)|Acc (%)|
> |:-:|:-:|:-:|:-:|
> |DeiT-S|eTPS|3.0|79.7
> ||dTPS*||80.1
> ||PPT (Ours)|2.9|79.8
>
> It should be noted that the dTPS* is fine-tuned under 100 epochs as mentioned in the paper, while the eTPS and our method both are fine-tuned under 30 epochs. We can find that our method achieves better performance than eTPS on the trade-off between FLOPs and accuracy.
>
> > **"Lack of experiments of hybrid ViT such as PVT..."**
>
> As we mentioned in the conclusion section, the classification token plays an important role in our framework, so we cannot apply our method to PVT directly. However, we are committed to addressing this limitation and finding a solution in our future research endeavors. We sincerely appreciate your suggestions.
>
> > **"I think the author should make the "adaptive" point more convincing, such as comparing other non-adaptive pruning and pooling methods (pruning and pooling at the same time), or comparing the naive non-adaptive baseline (pruning only before layer x and pooling only after layer x, layer x is a fixed parameter such as half or two-thirds of the depth)."**
>
> We highly appreciate your viewpoints, and in Section 4.2 of our paper, titled "**Effectiveness of techniques integration (Figure 8)**" and "**Different policy selection mechanisms (Table 3)**", we have discussed and provided further evidence for the two approaches you mentioned. These discussions aim to demonstrate the effectiveness and convincing of the "adaptive" strategies we proposed.
>
> > **"I think more metrics other than the "variance" of the distribution of importance score of image tokens should be considered. Since variance is shift-invariant and absolute value of the score is not considered. The absolute value of the score distribution should be somehow helpful to the choice of pooling or merging."**
>
> Thanks for your suggestion, we have explored **different metrics for redundancy discrimination** in Section C of Appendix due to the space limitations. Specifically, we have explored another metric, ***the average of token similarity***, which also reflects the redundancy level of tokens in different layers. And we find its performance is inferior to our chosen metric (The data placement in Table 7 is filled incorrectly, and we will refine the typo in refined version).
>
> > **"See weakness 3, how to make this method work on hybrid ViT(including regular spatial conv) and what's the improvement?"**
> > **"How to make this method work on dense prediction tasks such as segmentation and what's the performance?"**
>
> Thank you for your suggestions. Applying our method to models without class token is indeed one limitation of our method, as we also discuss in Page **conclusion**. We will leave it for future work.
>
>
> > **"Is it a possible case where pooling at the early stage and pruning at the final stage?**
> > **What's the performance if the adaptive policy is reversed? (change pooling to pruning and pruning to pooling)**
>
> As shown in Section 4.2 of our paper, titled "**Different policy selection mechanisms (Table 3)**",  we have explored the inversion of our policy decision, wherein pruning was performed instead of pooling, and vice versa. This significantly reduces the accuracy of the model, further reinforces our claims that token pooling techniques are typically applied in shallow layers, while token pruning techniques are preferably employed in deeper layers.
>
> We hope that our response adequately addresses your concerns. We genuinely appreciate your valuable discussion and suggestions, and we will refine the relevant content in the revised version accordingly.

---

> > ### Comment · Reviewer_b5bi · 2023-11-14
> >
> > Thank you for your response. The detailed explanation provided indeed reinforces the adaptive argument.
> >
> > I appreciate the thoughtful approach taken to address inattentive redundancy and duplicative redundancy across different layers, as well as the commendable off-the-shelf performance even without fine-tuning in this study. However, several efficient Vision Transformer (ViT) approaches[1][2] have been able to successfully balance high performance and compatibility for hybrid ViTs, which unfortunately makes the current work appear less impactful by comparison. Additionally, I concur with reviewer Egiv's assessment regarding the limited novelty present in the research lines involving both pruning and pooling for ViTs.
> >
> > In conclusion, my rating remains at 5: marginally below the acceptance threshold. I would suggest that the author strive to achieve a higher off-the-shelf performance to help distinguish this work from others.
> >
> > [1] TPS: https://arxiv.org/abs/2304.10716
> > [2] TOME: https://arxiv.org/abs/2210.09461

---

### Official Review · Reviewer_Egiv · 2023-10-28

**Soundness:** 3 good
**Presentation:** 3 good
**Contribution:** 2 fair
**Rating:** 5
**Confidence:** 4

**Summary:**

In this work, the authors propose to reduce computation load of ViT by adaptive token pruning and pooling. Base on the token distribution, they tackle inattentive and duplicative redundancy by an adaptively selection of token pruning or token pooling policy.

**Strengths:**

1 How to achieve accuracy-computation balance is a critical problem for ViT.

2 The propsoed method seems to be sound.

3 The paper is written well.

**Weaknesses:**

1 The novelty is relatively limited. The token pruning or token pooling has been widely investigated in the literature [DynamicViT, Evo-ViT, Self-Slim ViT, etc]. The adaptive design used in the paper is simply the combination of both  pruning and pooling without much insightful modifications.

2 The results are actually comparable with the state of the art methods, with a little improvement. As shown in Table 1, the proposed method is actually comparable to Evo-ViT,  in terms of all the DeiT model settings. Similarly, it shows the marginal performance improvement, compared with EViT-LV-S. All of these indicate that, the effectiveness of the proposed design is not quite convincing.

**Questions:**

Please see weakness for reference. Bascially, there are two main concerns.
1 The combination of pruning and pooling is not quite novel, considering that a number of similar compression methods have been developped in the literature.
2 The exepriments do not show the effectiveness of the proposed design, with a marignal improvement.

---

> ### Author Response · Authors · 2023-11-11
> **Response to reviewer Egiv**
>
> We sincerely appreciate your thoughtful comments, efforts, and time. We respond to each of your concerns and questions one-by-one in what follows:
>
> > **"The novelty is relatively limited. The token pruning or token pooling has been widely investigated in the literature [DynamicViT, Evo-ViT, Self-Slim ViT, etc]. The adaptive design used in the paper is simply the combination of both pruning and pooling without much insightful modifications."**
>
> We emphasize the novelty of our work in the following aspects:
>
> + The proposed method is the **heuristic** framework that acknowledges the complementary potential of token pruning and token merging techniques for effifient ViTs, i.e., token pruning for inattentive redundancy while token pooling for duplicative redundancy. The effectiveness of our approach is visually demonstrated in Figure 1 of the paper.
> + **Our method is not a simple combination of existing techniques**. As detailed in Section 3.4, our method is derived based on a closer look at the characteristics of token pruning and token merging techniques, as well as a thorough analysis of the importance scores in different layers.
> + Additionally, we heuristically design a **redundancy criterion**, i.e., the variance of the significance scores, to guide adaptive decision-making on prioritizing different token compression policies for various layers and instances, as shown in Figure 7.
>
> > **"The results are actually comparable with the state of the art methods, with a little improvement. As shown in Table 1, the proposed method is actually comparable to Evo-ViT, in terms of all the DeiT model settings. Similarly, it shows the marginal performance improvement, compared with EViT-LV-S. All of these indicate that, the effectiveness of the proposed design is not quite convincing."**
>
> We followed the settings of many previous works and reported the accuracy in Table 1 based on the given FLOPs. However, in this comparatively high FLOPs scenario, various methods have already approached the accuracy of the original model. Although our method achieves state-of-the-art performance, the improvement does not appear to be significant.
>
> To provide a more comprehensive evaluation, we show the comparison between our method and other methods under different FLOPs (based on the DeiT-S as baseline) and provide corresponding data in the Figure 4.
>
> As analyzed in paper, the superiority of our method is demonstrated in various FLOPs and off-the-shelf (without fine-tuning) scenarios. Specifically, it includes three aspects:
>
> + At lower FLOPs (2.5G below), our method exhibits a noteworthy performance improvement of 0.7%-3.8% compared to other pruning methods.
> + At higher FLOPs (3.0G above), our method demonstrates superior performance compared to the baseline model, surpassing the capabilities of the comparison methods.
> + When used in an off-the-shelf (without fine-tuning) setting, PPT demonstrates significant improvement (0.1%-2.3%) compared to other methods. Notably, PPT also achieves competitive results with other fine-tuned models.
>
> We hope that our response adequately addresses your concerns. We genuinely appreciate your valuable discussion and suggestions, and we will refine the relevant content in the revised version accordingly.

---

### Official Review · Reviewer_SM9c · 2023-10-30

**Soundness:** 2 fair
**Presentation:** 3 good
**Contribution:** 2 fair
**Rating:** 3
**Confidence:** 5

**Summary:**

This paper proposes to accelerate by token pruning and pooling, and its experiments show some improvement on throughput.

**Strengths:**

The strength of this paper is easy following. The figures help a lot for understanding the story. The experiments are good even though not better compared with some  recent papers.

**Weaknesses:**

The weakness of this paper are as follows:
1. The inference is hard to implement to be really act as what the authors have claimed. I do not think the throughputs are experimental numbers but theoretical numbers.  This is the common issue for adaptive token pruning methods, such as A-ViT. I think in practical this paper is useless, not only not decreasing the real computation cost but increasing the cost. I do not think the codes would be released for inference.
2. The results are not promising. In recent CVPR2023, there are several new works that achieved better results than this one. For example, Making Vision Transformers Efficient From a Token Sparsification View.
3. Further finetuning based on pretrained models is not fair comparisons.

**Questions:**

1. Can you share the inference codes?
2. Can you share the codes for throughputs calculation?

---

> ### Author Response · Authors · 2023-11-11
> **Response to reviewer SM9c**
>
> We sincerely appreciate your thoughtful comments, efforts, and time. We respond to each of your concerns and questions one-by-one in what follows:
>
> > **"The inference is hard to implement to be really act as what the authors have claimed. I do not think the throughputs are experimental numbers but theoretical numbers. This is the common issue for adaptive token pruning methods, such as A-ViT. I think in practical this paper is useless, not only not decreasing the real computation cost but increasing the cost. I do not think the codes would be released for inference."**
>
> Compared to A-ViT, which employs dynamic compression ratios based on different images, our approach utilizes a compression strategy that is more conducive to batch processing by progressively reducing the Top-K redundant tokens in each stage. This strategy significantly reduces the computational workload with only a minimal amount of additional computations required.
>
> As we mentioned in paper, **we measure the throughput of the models on a single NVIDIA V100 GPU with batch size fixed to 256 and we indeed decrease the real computation cost. The inference codes with throughputs calculation are updated in Supplementary Material. You can check it by running the eval.sh** (dependencies refer to requirements.txt).
>
> > **"The results are not promising. In recent CVPR2023, there are several new works that achieved better results than this one. For example, Making Vision Transformers Efficient From a Token Sparsification View."**
>
> Thank you for the complement to our comparative methods, which is a great work. In comparison to the method you mentioned, our method excels in that it does not require additional trainable parameters and can achieve impressive results even in **off-the-shelf** scenarios, where no customization or fine-tuning is required. We believe that this characteristic is particularly beneficial for huge models and scenarios with limited computational resources.
>
> > **"Further finetuning based on pretrained models is not fair comparisons."**
>
> Fine-tuning based on pretrained models is a widely adopted practice in this field. To ensure a fair comparison, we applied **the same fine-tuning strategy** as the previous methods, which include DynamicViT, EViT, PS-ViT, ATS, and others.
>
> We hope that our response adequately addresses your concerns. We genuinely appreciate your valuable discussion and suggestions, and we will refine the relevant content in the revised version accordingly.

---

### Official Review · Reviewer_zbfa · 2023-11-01

**Soundness:** 3 good
**Presentation:** 3 good
**Contribution:** 3 good
**Rating:** 6
**Confidence:** 4

**Summary:**

This paper proposed to integrate both token pruning and token pooling techniques into vision transformers to perform model compression, and input-adaptively deciding different token compression policies for different layers, without additional trainable parameters. The compression results look promising.

**Strengths:**

1. The proposed method is simple yet effective, without additional trainable parameters, and can be easily incorporated into the standard transformer block.
2. The results demonstrate a better accuracy-compression ratio trade-off than the previous methods.

**Weaknesses:**

1. Missing a pretty relevant reference: Unified Vision Transformer Compression.  ICLR 2022 Please cite and compare with it on Deit-S, Deit-B, and Deit-Tiny.
2. The backbones used in the paper are Deit and LV-VIT. How about the Swin-Transformer, which also already has a patch merging module for each stage of blocks? I am curious about the generalization of the proposed mechanism in this kind of structure and its performance.

**Questions:**

The target compression ratio seems to be hard to directly and accurately map to each layer's reduction percentage contributed by token pooling and token pruning. How to balance the K threshold in pruning and pooling and how to choose the decision threshold to match a user-given compression ratio? Please clarify it in more detail.

---

> ### Author Response · Authors · 2023-11-11
> **Response to reviewer zbfa**
>
> We sincerely appreciate your thoughtful comments, efforts, and time. We respond to each of your concerns and questions one-by-one in what follows:
>
> > **"Missing a pretty relevant reference: Unified Vision Transformer Compression. ICLR 2022 Please cite and compare with it on Deit-S, Deit-B, and Deit-Tiny."**
>
> In our paper, we primarily compare acceleration techniques for Vision Transformers (ViTs) that focus on ***token-level*** compression. However, as you mentioned, the UVC method you referred to primarily compresses the ***model parameters***. We appreciate your reminder and would like to cite it in paper and incorporate a comparison of its performance as follows:
>
> |Model|Method|FLOPs (G)|Acc (%)|
> |:-:|:-:|:-:|:-:|
> |DeiT-Tiny|UVC|0.7|71.8
> ||PPT (Ours)|0.8|72.1
> |DeiT-S|UVC|2.7|79.4
> ||PPT (Ours)|2.7|79.6
> |DeiT-B|UVC|8.0|80.6
> ||PPT (Ours)|11.6|81.4
>
> > **"The backbones used in the paper are Deit and LV-VIT. How about the Swin-Transformer, which also already has a patch merging module for each stage of blocks? I am curious about the generalization of the proposed mechanism in this kind of structure and its performance."**
>
> In the Swin-Transformer, the patch merging module is designed to reduce the resolution, similar to pooling, by merging local features in a regular manner. However, our pruning and pooling techniques are actually instance-aware to reduce the computation cost, and these two approaches are not contradictory. In our future work, we will investigate experiments combining our methods with the Swin-Transformer. We sincerely appreciate your suggestions.
>
> > **"The target compression ratio seems to be hard to directly and accurately map to each layer's reduction percentage contributed by token pooling and token pruning. How to balance the K threshold in pruning and pooling and how to choose the decision threshold to match a user-given compression ratio? Please clarify it in more detail."**
>
> As illustrated in Figure 4, we showcase the performance of our method across various FLOPs by adjusting the hyperparameter *K*. For more comprehensive details, please refer to Section B in the appendix, where we provide specific settings and explain how to select the appropriate *K* to achieve a desired compression ratio. Additionally, in Section 4.2 of our paper, we delve into a thorough discussion on the **impact of the decision threshold** and present Figure 9, which outlines the process for determining the optimal threshold value.
>
> We hope that our response adequately addresses your concerns. We genuinely appreciate your valuable discussion and suggestions, and we will refine the relevant content in the revised version accordingly.